# Diffusion-LLM Provides Ultra-Long-Term Time Series Forecasting with Probabilistic Alignment

## Abstract

Time series forecasting is a fundamental task in machine learning. Recently, Large Language Models (LLMs) have gained attention for this task due to their strong generalization capabilities, particularly in recognizing patterns and performing complex reasoning across diverse data modalities. Apart from having the architecture suitable for long-context learning, LLMs are an interesting option also because of their few-shot and zero-shot transfer learning capability, making it possible to use pretrained frozen LLMs directly for time series forecasting. However, challenges remain in adapting LLMs to multimodal tasks: they often lack a calibrated understanding of probabilistic structure in non-text modalities and struggle with aligning heterogeneous representations. To address these limitations, we propose `Diffusion-LLM`, a novel framework that integrates a conditional diffusion model into an LLM-based forecasting pipeline. This joint setup enables the model to learn the conditional distribution of future time series trajectories while reinforcing semantic alignment in the shared latent space. We evaluate `Diffusion-LLM` on six standard long-term forecasting benchmarks, including ETT, Weather, and ECL datasets. Our approach consistently outperforms existing LLM-based baseline, achieving substantial gains in ultra-long-term and few-shot forecasting tasks, while demonstrating the effectiveness of distribution-aware regularization for enhancing the robustness and generalization of time series LLMs.

## 1 Introduction

Time series forecasting has relevant applications in domains such as energy systems (Uremović et al., 2023; Chou & Tran, 2018), healthcare monitoring (Morid et al., 2023), climate science (Karevan & Suykens, 2020), and supply chain management (Pacella & Papadia, 2021). While most models are optimized for short-term to long-term horizons, many real-world scenarios like energy sector, climate science, vehicle industry etc. require accurate predictions far beyond this range (Wang et al., 2023). For example, prediction needs in energy demand forecasting can range anywhere, starting from a few hours, days or weeks extending into months and even years. Such *ultra-long-term* forecasting tasks, beyond thousands or more steps ahead, must often rely on limited historical data, making them especially challenging but important for strategic decision-making and long-term risk assessment. Battery lifetime prediction from early aging data is another example of this requirement (Li et al., 2024).

Leveraging pretrained LLMs has become an increasingly promising approach for time series forecasting, thanks to their strong pattern recognition and reasoning abilities, and flexible integration options for existing pipelines. Notably, LLMs exhibit powerful inductive capabilities even without task-specific fine-tuning. Gruver et al. (2023) show that LLMs can achieve impressive zero-shot performance across a variety of tasks.

However, finetuning LLMs is expensive and the large parameter capacity in transformer-based solutions can lead to overfitting for time series data (Zeng et al., 2023). Thus, applying LLMs to time series data introduces unique challenges. Unlike natural language, which is governed by semantic and syntactic structures, time series data is characterized by temporal dependencies and often lacks the rich contextual cues present in text. This domain mismatch makes it difficult to align time

series and language representations within a shared embedding space, leading to degraded performance. In multimodal applications, lack of sufficient multimodal alignment is also the main reason for hallucinations in LLMs (Shukor & Cord, 2024).

Pretrained LLMs excel at modeling probabilistic relationships in the text domain, as their attention mechanisms are inherently optimized to predict the most likely next token based on grammatical structure and semantic context. However, their ability to capture the data distribution in time series is limited without additional fine-tuning or specialized learning frameworks. This limitation becomes more pronounced in LLM-based time series forecasting models trained with Mean Squared Error (MSE) loss, which tend to regress toward the mean. As a result, these models struggle to represent the full distribution of possible futures, particularly in non-periodic or noisy datasets. While LLMs can effectively detect periodic patterns, they are less capable of modeling irregular or highly variable time series (Tang et al., 2025). Furthermore, LLMs have limited capacity to generate coherent and accurate time series over extended horizons. During generation, predictions are based on both the model's learned context and the partially generated output. As sequences grow longer, attention mechanisms increasingly focus on recent tokens, leading to reduced awareness of the broader context (Shi et al., 2023). This shift toward localized attention results in overconfidence, where the model prioritizes nearby outputs and underestimates uncertainty (Huang et al., 2025). In the context of time series forecasting, this behavior can cause performance to degrade progressively with longer prediction windows.

To jointly address these limitations, we introduce a Denoising Diffusion Probabilistic Model (DDPM) (Ho et al., 2020) into the forecasting framework. DDPMs are a class of generative models that can estimate complex data distributions through a gradual denoising process. They have shown remarkable success in domains such as image synthesis and inpainting, where modeling conditional probabilities is key. In our setting, the DDPM estimates the probability distribution of the forecasting window conditioned on the lookback window. Input adaptation has been shown to be feasible through the tokenization and embedding scheme introduced in TimeLLM (Jin et al., 2024), referred to as *reprogramming*, which encodes both the lookback and forecasting windows as sequences of word-like prototypes in a shared latent space. The DDPM is jointly trained with the LLM framework to estimate the conditional distribution of the encoded forecasting window given the encoded input. This provides a distribution-aware training signal that acts as a regularizer, enhancing the LLM's ability to model uncertainty. This dual objective training enables both robustness and predictive performance of our proposed `Diffusion-LLM` method. Our main contributions in this work can be summarized as follows:

- We introduce the novel idea of using generative models like DDPMs as an implicit regularizer for multimodal LLMs. This enables joint alignment between textual and temporal representations while modeling their shared distribution in a unified embedding space.

- We propose and implement `Diffusion-LLM`, a new framework that estimates the probability distribution of reprogrammed time series patches within the multimodal embedding space, enhancing an LLM's ability to reason over temporal data.

- We demonstrate that our framework improves ultra-long-term and few-shot forecasting performance across multiple standard benchmarks.

## 2 RELATED WORK

### 2.1 LLM IN TIME SERIES FORECASTING:

Recent research has explored various strategies to adapt LLMs for time series analysis. Prompting-based methods treat time series as raw text and directly feed them into LLMs using handcrafted templates but suffer from loss of semantics due to modality difference (Xue & Salim, 2023; Gruver et al., 2023). Quantization approaches convert time series into discrete tokens using techniques like VQ-VAE or K-means clustering and may require two-stage training (Talukder et al., 2024; Yu et al., 2023). Vision-as-bridge methods transform time series into visual representations (*e.g.*, line plots or spectrograms) and use vision-language models to interpret them. While effective in some domains, this approach depends heavily on the availability of paired visual data and may not generalize well (Wimmer & Rekabsaz, 2023). Tool-based methods use LLMs to generate auxiliary

tools such as code or API calls for downstream tasks (Qin et al., 2024). These often require complex integration and are less suited for end-to-end forecasting.

Alternatively, alignment-based methods like Zhang et al. (2024) aim to learn an encoding of time series and align the encoded time series to the semantic space of language models, enabling more robust and semantically meaningful interactions. These can be broadly divided into two subcategories.

- **Contrastive alignment:** Methods like ETP (Liu et al., 2024), TEST (Sun et al., 2024), and TENT (Zhou et al., 2023b) use contrastive learning to align time series and text embeddings by maximizing similarity between paired representations. These approaches are effective when multimodal data is available, such as for aligning ECG signals with clinical reports or IoT sensor data with activity descriptions.

- **LLM-backbone alignment:** Works such as GPT4TS (Zhou et al., 2023a), LLM4TS (Chang et al., 2025), and TimeLLM (Jin et al., 2024), directly feed reprogrammed time series embeddings into frozen or partially frozen LLMs. These models often use patching, decomposition, or domain-specific prompts to enhance alignment and perform better at activating the pretrained LLM's knowledge transfer and reasoning capabilities. GPT4TS freezes the self-attention layers of the LLM while fine-tuning as they contain a majority of the pretrained LLM's learned knowledge. LLM4TS (Chang et al., 2025) uses a two-stage process: first, an autoregressive approach to align pretrained LLM with patched time series and then Parameter-Efficient Fine-Tuning methods to selectively adjust a limited portion of the LLM parameters. TimeLLM (Jin et al., 2024) reprograms time series into token sequences that are aligned with LLM's text prototypes to resemble natural language, allowing LLMs to process them using their native architecture. Time-VLM (Zhong et al., 2025) goes one step further by adding image modality to a frozen VLM framework.

Our work builds upon this alignment-based paradigm and introduces a diffusion-based regularization mechanism to enhance distributional modeling. This is in contrast to more conservative methods like Benidis et al. (2022); Yang et al. (2025); Hyndman & Athanasopoulos (2018); Wen et al. (2018), which often focus on enhancing forecasting through multiscale input decomposition and predominantly linear models. These approaches can be effective for deterministic settings, but they do not address the probabilistic modeling of uncertainty, nor do they explore multimodal alignment or LLM-based reasoning.

## 2.2 DDPM in Time Series Forecasting:

Recent works have explored the use of DDPMs for time series forecasting, primarily by combining them with autoregressive backbones. These models typically generate future sequences in a denoising fashion. For instance, TimeGrad (Rasul et al., 2021) first injects noise to data at each predictive time point, and then denoises through a backward process conditioned on the encoded lookback window. The lookback window is encoded using hidden state from a Recurrent Neural Network (RNN) module like Long Short-Term Memory (LSTM) (Hochreiter & Schmidhuber, 1997). Score-Grad (Yan et al., 2021) uses a feature extraction method almost identical to TimeGrad but combines it with a conditional SDE-based score-matching module for the diffusion process. Contrary to these existing works, our work does not directly use DDPM as generative forecaster and instead focuses on using DDPM as an auxiliary learner to improve the robustness of LLM-based frameworks.

## 3 Methodology

Our proposed framework, `Diffusion-LLM`, enhances LLM-based time series forecasting by integrating a conditional DDPM as a regularization mechanism. The key idea is to estimate the conditional distribution of the forecasting window given the lookback window in a shared embedding space of text prototypes obtained by reprogramming the time series, thereby improving both the probabilsitc modeling and multimodal alignment capabilities of the LLM. An overview of the proposed `Diffusion-LLM` architecture during training is illustrated in Figure 1.

Our approach consists of three main components:

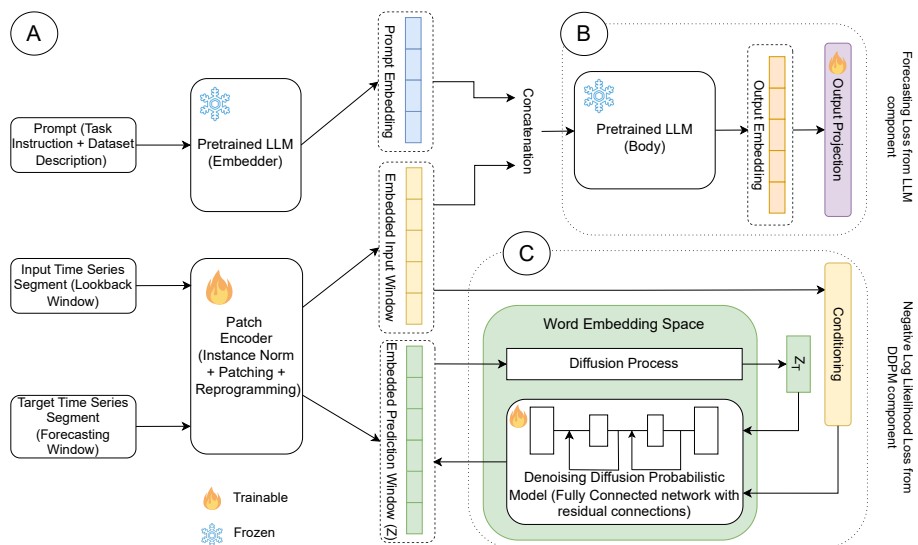

Figure 1: Training architecture of `Diffusion-LLM`. (A) The prompt, input, and target time series are reprogrammed into a shared token embedding space using a frozen LLM encoder and a trainable patch encoder. (B) The encoded input is used for direct forecasting via a frozen LLM output module. (C) A conditional DDPM is trained to model the distribution of the encoded target, conditioned on the input, by predicting the added noise. The final loss combines forecasting and diffusion-based regularization.

***A. Time Series Encoder (Reprogramming and Embedding):*** Following the reprogramming strategy introduced in TimeLLM by (Jin et al., 2024), we tokenize raw time series data and use an attention-based method to learn relevant text prototypes for different patches. The encoding and reprogramming mechanism is described in detail and illustrated in Figure 2b. During training, along with the lookback window $x$ encoded in TimeLLM, we also encode the forecasting window $y$ using the shared encoder $\phi_{\text{llmenc}}$, producing latent representations $z_x$ and $z_y$ respectively. Similar to the baseline, we use three parts in the prompt design: 1. Dataset details, 2. Task instruction, and 3. Statsitical information. For example, a sample prompt for Weather dataset is "Weather is recorded every 10 minutes for the 2020 whole year, which contains 21 meteorological indicators, such as air temperature, humidity, etc. Predict the next 2048 steps given the previous 512 steps information attached. The input has a minimum of .., a maximum of .., and a median of ... The overall trend is ... The top five lags are ..." We also retain the prompt embedding of TimeLLM but with a slight simplification of notations, we ignore the frozen prompt embedder in the equation and describe the patch encoder itself as $\phi_{\text{llmenc}}$,

$$z_x = \phi_{\text{llmenc}}(x), \quad z_y = \phi_{\text{llmenc}}(y). \tag{1}$$

The encoded time series serves as semantically meaningful tokens within the language model's embedding space, enabling effective processing by pretrained LLMs using their native architecture.

***B. Forecasting via LLM:*** The encoded input $z_x$ is passed to a frozen LLM-based output module $\phi_{\text{llmout}}$ that consists of the frozen LLM and an output projection layer and generates the predicted forecast $\hat{y}$:

$$\hat{y} = \phi_{\text{llmout}}(z_x). \tag{2}$$

The forecasting loss is computed using Mean Squared Error between the predicted and actual target values:

$$\mathcal{L}_{\text{forecast}} = \|y - \hat{y}\|^2. \tag{3}$$

This component leverages the pretrained reasoning and pattern recognition capabilities of LLMs while avoiding full finetuning, thus maintaining efficiency and generalization. As shown by Jin et al. (2024); Dombrowski et al. (2024), such model reprogramming approaches of frozen LLMs can be more efficient than parameter-efficient fine-tuning methods like QLoRA (Dettmers et al., 2023).

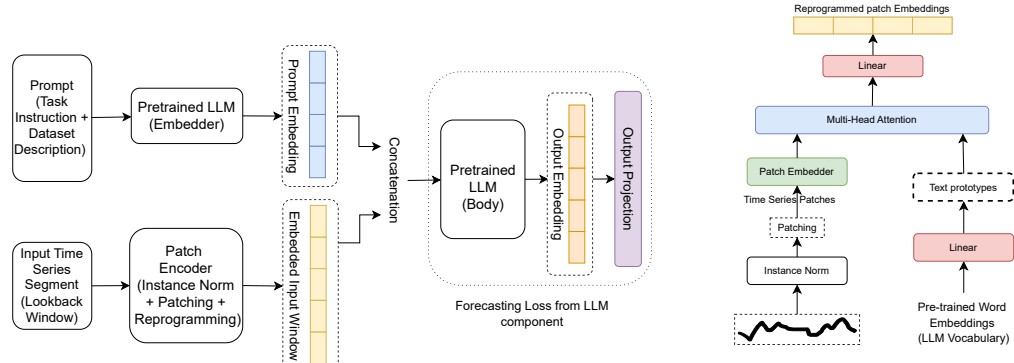

(a) Inference pipeline.

(b) Patch Encoding mechanism.

Figure 2: (a) The inference pipeline of `Diffusion-LLM`. Only the LLM modules are used to generate forecasts from new input data. (b) The patch Encoding mechanism in `Diffusion-LLM`. The inputs are the Time series window and the pre-trained word embeddings of the LLM (vocabulary). The time series is normalized and patched. For efficiency, only a selected few text prototypes are constructed through the linear layer in a learnable manner. The attention mechanism between the time series patches and the text prototypes helps the LLM to learn the relevant tokens or language cues (words or phrases like 'short up', 'steady down', 'periodic' etc.) for characterizing each patch in the token embedding space where the language model is pre-trained. The encoding parameters are trained in end-to-end manner as part of the whole framework.

***C. Distribution Regularization via DDPM:*** To improve the model's ability to capture token distribution of time series representation, we use a conditional DDPM. The objective for the DDPM is to learn the conditional distribution $p(z_y \mid z_x)$ through a denoising process. During training, noise is added to $z_y$ to produce a noisy version $\tilde{z}_y$, and the DDPM is trained to predict the noise $\epsilon$:

$$\tilde{z}_y \sim q(\tilde{z}_y \mid z_y, t), \tag{4}$$

$$\hat{\epsilon} = \epsilon_\theta(\tilde{z}_y, t, z_x), \tag{5}$$

$$\mathcal{L}_{\text{ddpm}} = \|\epsilon - \hat{\epsilon}\|^2. \tag{6}$$

As shown in Ho et al. (2020), this is equivalent to learning the conditional probability distribution,

$$\mathcal{L}_{\text{ddpm}} = -\log p_\theta(z_y \mid z_x). \tag{7}$$

While the DDPM component can be interpreted as a regularizer, our framework also broadly fits within the paradigm of multi-task learning that can be cast as multi-objective optimization (Sener & Koltun, 2018) with the model being jointly optimized for both forecasting and distribution estimation,

$$\mathcal{L}_{\text{joint}} = \mathcal{L}_{\text{forecast}} + \lambda \cdot \mathcal{L}_{\text{ddpm}}. \tag{8}$$

This dual-objective setup allows the DDPM to act as a probabilistic constraint and also as an auxiliary learner that enriches the shared embedding space through semantic alignment. Moreover, the DDPM regularization is agnostic to the exact alignment approach and can be integrated with existing methods with minimal code changes similar to the enhancement shown here on TimeLLM (Jin et al., 2024).

Thus, we jointly optimize the LLM parameters $\phi_{\text{llmenc}}$ and $\phi_{\text{llmout}}$ and the DDPM parameters $\theta_{\text{ddpm}}$. The complete training procedure is detailed in Algorithm 1.

---

**Algorithm 1 Diffusion-LLM Training**

---

**Require:** Time series dataset $\mathcal{D} = \{(x, y)\}$, LLM encoder module $\phi_{\text{llmenc}}$, LLM output module $\phi_{\text{llmout}}$, DDPM model $\theta_{\text{ddpm}}$, regularization weight $\lambda$.

Initialize parameters of $\phi_{\text{llmenc}}, \phi_{\text{llmout}}, \theta_{\text{ddpm}}$.

**for** each training iteration **do**

    Sample a batch $\mathcal{B} = \{(x_i, y_i)\}$ from $\mathcal{D}$

    **for** each $(x, y)$ in $\mathcal{B}$ **do**

        **1. Encode input and target windows**

          (a) $z_x \leftarrow \phi_{\text{llmenc}}(x)$, (b) $z_y \leftarrow \phi_{\text{llmenc}}(y)$

        **2. Forecasting prediction and loss**

          (a) $\hat{y} \leftarrow \phi_{\text{llmout}}(z_x)$, (b) $\mathcal{L}_{\text{forecast}} \leftarrow \|y - \hat{y}\|^2$

        **3. DDPM loss**

          (a) Sample noise $\epsilon \sim \mathcal{N}(0, I)$ and timestep $t \sim \text{Uniform}(1, T)$

          (b) Noised sample: $\tilde{z}_y = \sqrt{\bar{\alpha}_t} z_y + \sqrt{1 - \bar{\alpha}_t} \cdot \epsilon$

          (c) Predict noise: $\hat{\epsilon} \leftarrow \theta_{\text{ddpm}}(\tilde{z}_y, t, z_x)$

          (d) $\mathcal{L}_{\text{ddpm}} \leftarrow \|\epsilon - \hat{\epsilon}\|^2$

        **4. Combine losses**

          (a) $\mathcal{L}_{\text{joint}} \leftarrow \mathcal{L}_{\text{forecast}} + \lambda \cdot \mathcal{L}_{\text{ddpm}}$

    **end for**

    Update $\phi_{\text{llmenc}}, \phi_{\text{llmout}}, \theta_{\text{ddpm}}$ using gradients of $\mathcal{L}_{\text{joint}}$

**end for**

---

During inference, only the LLM modules are used to generated forecasts from new input data (Figure 2a). The inference steps are formally defined in Algorithm 2.

---

**Algorithm 2 Diffusion-LLM Inference**

---

**Require:** Input time series $x$, trained encoder $\phi_{\text{llmenc}}$, trained output module $\phi_{\text{llmout}}$

    **1. Encode the input window**

      $z_x \leftarrow \phi_{\text{llmenc}}(x)$

    **2. Generate forecast**

      $\hat{y} \leftarrow \phi_{\text{llmout}}(z_x)$

    **return** $\hat{y}$ as the predicted forecasting window

---

## 4 EXPERIMENTS AND RESULTS

### A. Model Architecture

We use the 7B variant of LLaMA (Touvron et al., 2023) as the backbone LLM in all our experiments. For the diffusion component, we adopt a lightweight Denoising Diffusion Probabilistic Model (DDPM) implemented as a stack of fully connected layers with skip connections. All experiments are conducted on NVIDIA A100 and H100 GPUs.

### B. Long-Term Forecasting

We evaluate `Diffusion-LLM` on six widely used long-term forecasting benchmarks: ETTh1, ETTh2, ETTm1, ETTm2 (ETT dataset from Zhou et al. (2021)), Weather, and Electricity (ECL) (both from Wu et al. (2023)). Lookback window of length 512 and forecasting horizons of $\{96, 192, 336, 720\}$ are used. ILI dataset (Wu et al., 2023) was considered but its shorter standard forecasting window of $H \in \{24, 36, 48, 60\}$ and unavailability of enough data for ultra-long forecasting make it unsuitable for our evaluation. As we present our method as a simple add-on improvement over existing LLM-based methods, we show competitive results with the existing benchmarks (Table 3) but more importantly, make exhaustive comparison against the baseline method of TimeLLM and report the results along with standard deviation for Mean Squared Error (MSE) and Mean Absolute Error (MAE) evaluation metrics (Table 1 and Table 2). `Diffusion-LLM` yields similar results as TimeLLM for this task.

### C. Ultra-Long-Term Forecasting

| Dataset | Long-term | | | | Ultra-long-term | | | |
| --- | --- | --- | --- | --- | --- | --- | --- | --- |
| | TimeLLM | | Diffusion-LLM (Ours) | | TimeLLM | | Diffusion-LLM (Ours) | |
| | MSE | MAE | MSE | MAE | MSE | MAE | MSE | MAE |
| ETTh1 | $0.449_{\pm0.025}$ | $0.457_{\pm0.015}$ | $\mathbf{0.427}_{\pm0.004}$ | $\mathbf{0.446}_{\pm0.010}$ | $0.758_{\pm0.018}$ | $0.600_{\pm0.011}$ | $\mathbf{0.612}_{\pm0.011}$ | $\mathbf{0.558}_{\pm0.004}$ |
| ETTh2 | $\mathbf{0.373}_{\pm0.009}$ | $\mathbf{0.409}_{\pm0.006}$ | $0.387_{\pm0.003}$ | $0.425_{\pm0.002}$ | $0.589_{\pm0.013}$ | $0.543_{\pm0.007}$ | $\mathbf{0.522}_{\pm0.009}$ | $\mathbf{0.512}_{\pm0.004}$ |
| ETTm1 | $0.381_{\pm0.008}$ | $0.406_{\pm0.006}$ | $\mathbf{0.376}_{\pm0.004}$ | $\mathbf{0.399}_{\pm0.002}$ | $0.484_{\pm0.009}$ | $0.472_{\pm0.012}$ | $\mathbf{0.465}_{\pm0.001}$ | $\mathbf{0.452}_{\pm0.001}$ |
| ETTm2 | $\mathbf{0.271}_{\pm0.003}$ | $\mathbf{0.330}_{\pm0.003}$ | $0.334_{\pm0.003}$ | $0.369_{\pm0.001}$ | $\mathbf{0.410}_{\pm0.020}$ | $\mathbf{0.425}_{\pm0.014}$ | $0.422_{\pm0.008}$ | $0.436_{\pm0.004}$ |
| Weather | $\mathbf{0.259}_{\pm0.019}$ | $\mathbf{0.288}_{\pm0.017}$ | $0.304_{\pm0.001}$ | $0.329_{\pm0.001}$ | $0.424_{\pm0.008}$ | $0.401_{\pm0.004}$ | $\mathbf{0.407}_{\pm0.001}$ | $\mathbf{0.394}_{\pm0.001}$ |
| ECL | $\mathbf{0.171}_{\pm0.002}$ | $\mathbf{0.277}_{\pm0.003}$ | $0.200_{\pm0.004}$ | $0.303_{\pm0.002}$ | $\mathbf{0.272}_{\pm0.001}$ | $\mathbf{0.356}_{\pm0.000}$ | $0.297_{\pm0.005}$ | $0.376_{\pm0.004}$ |

Table 1: Comparison of TimeLLM (Jin et al., 2024) and `Diffusion-LLM` across long-term and ultra-long-term forecasting tasks on standard time series benchmarks. **Long-term** forecasting results are averaged over four prediction horizons: $H \in \{96, 192, 336, 720\}$, using an input sequence length of 512. **Ultra-long-term** refers to the average performance over extended horizons $H \in \{1024, 2048\}$, which pose greater challenges due to increased uncertainty and weaker temporal correlations. Each cell reports the MSE and MAE and their standard deviations across multiple runs. Lower values indicate better performance and best results are indicated in bold. '-' means that data quantity is not sufficient to constitute a meaningful training set. `Diffusion-LLM` outperforms TimeLLM on 4/6 datasets for ultra-long-term forecasting. The magnitude of improvement in performance is remarkable for smaller datasets like ETTh1 and ETTh2, underlining the generalization ability of our method.

| Dataset | Few-shot (10%) long-term | | | | Few-shot (10%) ultra-long-term | | | | Few-shot (5%) long-term | | | | Few-shot (5%) ultra-long-term | | | |
| --- | --- | --- | --- | --- | --- | --- | --- | --- | --- | --- | --- | --- | --- | --- | --- | --- |
| | TimeLLM | | Diffusion-LLM (Ours) | | TimeLLM | | Diffusion-LLM (Ours) | | TimeLLM | | Diffusion-LLM (Ours) | | TimeLLM | | Diffusion-LLM (Ours) | |
| | MSE | MAE | MSE | MAE | MSE | MAE | MSE | MAE | MSE | MAE | MSE | MAE | MSE | MAE | MSE | MAE |
| ETTh1 | $0.834_{\pm0.073}$ | $0.614_{\pm0.022}$ | $\mathbf{0.662}_{\pm0.004}$ | $\mathbf{0.564}_{\pm0.001}$ | - | - | - | - | $0.988_{\pm0.066}$ | $0.662_{\pm0.021}$ | $\mathbf{0.728}_{\pm0.029}$ | $\mathbf{0.582}_{\pm0.013}$ | - | - | - | - |
| ETTh2 | $0.422_{\pm0.009}$ | $0.443_{\pm0.005}$ | $\mathbf{0.398}_{\pm0.003}$ | $\mathbf{0.432}_{\pm0.002}$ | - | - | - | - | $0.415_{\pm0.011}$ | $0.435_{\pm0.008}$ | $\mathbf{0.392}_{\pm0.003}$ | $\mathbf{0.428}_{\pm0.003}$ | - | - | - | - |
| ETTm1 | $0.504_{\pm0.001}$ | $\mathbf{0.462}_{\pm0.003}$ | $\mathbf{0.502}_{\pm0.029}$ | $0.464_{\pm0.014}$ | $1.056_{\pm0.101}$ | $0.691_{\pm0.036}$ | $\mathbf{0.660}_{\pm0.062}$ | $\mathbf{0.550}_{\pm0.026}$ | $0.600_{\pm0.011}$ | $0.515_{\pm0.006}$ | $\mathbf{0.528}_{\pm0.014}$ | $\mathbf{0.480}_{\pm0.005}$ | $0.924_{\pm0.032}$ | $0.666_{\pm0.011}$ | $\mathbf{0.628}_{\pm0.003}$ | $\mathbf{0.536}_{\pm0.001}$ |
| ETTm2 | $0.327_{\pm0.017}$ | $\mathbf{0.361}_{\pm0.008}$ | $0.336_{\pm0.003}$ | $0.370_{\pm0.003}$ | $0.582_{\pm0.022}$ | $0.506_{\pm0.003}$ | $\mathbf{0.442}_{\pm0.000}$ | $\mathbf{0.447}_{\pm0.000}$ | $0.330_{\pm0.001}$ | $\mathbf{0.367}_{\pm0.003}$ | $0.346_{\pm0.001}$ | $0.381_{\pm0.003}$ | $0.522_{\pm0.018}$ | $0.480_{\pm0.004}$ | $\mathbf{0.450}_{\pm0.004}$ | $\mathbf{0.444}_{\pm0.006}$ |
| Weather | $\mathbf{0.256}_{\pm0.000}$ | $\mathbf{0.291}_{\pm0.002}$ | $0.319_{\pm0.008}$ | $0.340_{\pm0.004}$ | $0.480_{\pm0.005}$ | $0.430_{\pm0.002}$ | $\mathbf{0.428}_{\pm0.003}$ | $\mathbf{0.406}_{\pm0.000}$ | $0.304_{\pm0.006}$ | $\mathbf{0.326}_{\pm0.001}$ | $0.329_{\pm0.005}$ | $0.347_{\pm0.003}$ | $0.477_{\pm0.007}$ | $0.434_{\pm0.004}$ | $\mathbf{0.424}_{\pm0.006}$ | $\mathbf{0.406}_{\pm0.000}$ |
| ECL | $\mathbf{0.190}_{\pm0.000}$ | $\mathbf{0.288}_{\pm0.001}$ | $0.197_{\pm0.000}$ | $0.294_{\pm0.000}$ | $0.292_{\pm0.000}$ | $0.367_{\pm0.000}$ | $\mathbf{0.281}_{\pm0.001}$ | $\mathbf{0.358}_{\pm0.004}$ | $\mathbf{0.192}_{\pm0.000}$ | $\mathbf{0.289}_{\pm0.001}$ | $0.201_{\pm0.003}$ | $0.298_{\pm0.000}$ | - | - | - | - |

Table 2: Comparison of TimeLLM (Jin et al., 2024) and `Diffusion-LLM` across few-shot long and ultra-long-term forecasting on standard time series benchmarks. **Few-shot (10%)** and **Few-shot (5%)** indicate training with only 10% and 5% of the available training data, respectively, to evaluate generalization under data scarcity. Other details are according to the protocol in Table 1. '-' means that data quantity is not sufficient to constitute a meaningful training set. Our method consistently outperforms TimeLLM on few-shot ultra-long forecasting tasks.

.

Ultra-long-term forecasting is particularly challenging due to increased uncertainty and weaker correlations with recent history. We evaluated this setting using the same datasets but focus on longer prediction horizons ($\{1024, 2048\}$). `Diffusion-LLM` outperforms TimeLLM in this regime in multiple datasets, demonstrating the benefit of modeling the full conditional distribution of the target window (Table 1). For relatively smaller datasets like ETTh1 and ETTh2, the benefits are particularly remarkable, with MSE reduction of 19.26%, and 11.38%, respectively.

### D. Few-Shot Forecasting (10% and 5%)

To evaluate few-shot generalization, we train both models using only 10% of the available training data. We follow the same setup as TimeLLM and report results on all eight datasets. `Diffusion-LLM` performs noticeably better than TimeLLM (Table 2), showing improvements across all ultra-long-forecasting scenarios. On ETTh1, our method outperforms TimeLLM by 20.62% even for long-term forecasting. This indicates that the diffusion-based regularization can enhance generalization greatly in low-data regimes, without requiring any fine-tuning of the LLM backbone.

### E. Few-Shot Forecasting (5%)

In the more extreme 5% few-shot setting, `Diffusion-LLM` shows clearer advantages over TimeLLM (Table 2). On ETTh1, this corresponds to a 25.79% improvement for long-term forecasting. This highlights the benefit of distribution-aware learning when data is scarce and uncertainty is high.

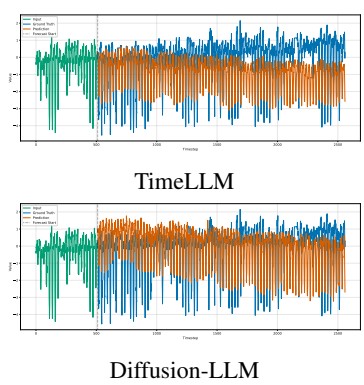

TimeLLM

Diffusion-LLM

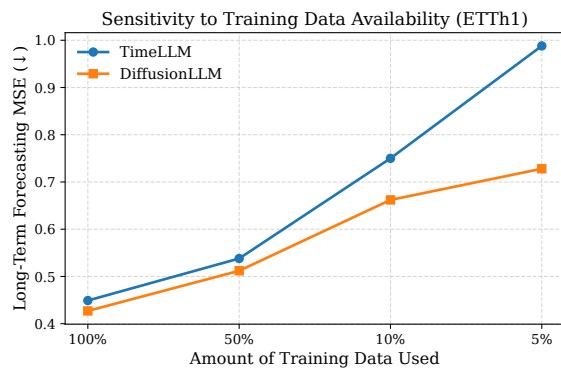

**(a)** Forecasting visualization

**(b)** Performance comparison under data-scarcity

Figure 3: (a) Visualization of ultra-long-term forecasting on ETTh1 dataset sample for 512 lookback and 2048 forecast window. TimeLLM shows considerable deviation from the ground truth in the later parts of forecast window while `Diffusion-LLM` shows consistent performance over the whole window. (b) Comparison of long-term forecasting performance of TimeLLM and `Diffusion-LLM` shows slower performance degradation and more robustness for `Diffusion-LLM` in data-scarcity scenarios. Protocol is same as Table 1.

Our results show that even though an LLM alone can often capture short-term patterns well, for very long-term forecasting with more uncertainty and variability and weaker direct correlation with recent past, modelling the complete distribution is more effective. Integrating the DDPM helps the encoder to learn richer representations in the embedding space. Because of Diffusion model's ability to model probability distributions, DDPM-based distribution regularization is most beneficial when uncertainty is high. This design introduces an optimization trade-off: while it improves robustness in high-uncertainty regimes, it can slightly reduce point prediction accuracy for shorter horizons. The benefits of distribution regularization in handling uncertainty is supported by the empirical evidence in tables 1 and 2 that Diffusion-LLM shows larger improvements on the most challenging datasets which exhibit higher baseline MSE (*e.g.* ETTh1, ETTm1). Even for easier datasets with lower MSE (*e.g.* Weather), as forecasting horizons lengthen and training data becomes scarce gradually, the advantages of Diffusion-LLM show, outperforming the baseline under these high-uncertainty conditions. The strong robustness of `Diffusion-LLM` under limited training data is further illustrated in Figure 3 as its performance degrades considerably less with increasing data scarcity compared to TimeLLM.

## 5 MODEL ANALYSIS

Here, we present the analysis from the experiments that serve as ablation studies and highlight the design decisions that contributed to the model's performance. The empirical results are presented in Table 5) in the appendix section A.3.

**Architectural Variants:** We experimented with two primary architectures for the DDPM: a 1D version with standard U-Net (Ronneberger et al., 2015) and a fully connected network with skip connections. Despite the expressive capacity of U-Net, we observed that the simpler fully connected architecture with fewer parameters yielded comparable or better performance, suggesting that overparameterization is not necessary.

**Conditioning Strategies:** In the default setup, the DDPM receives the concatenated embeddings of both the prompt and the reprogrammed time series. We tested conditioning via two methods: concatenation with input and timestep embeddings and attention mechanisms within the denoising process. Our findings indicate that simpler conditioning like direct concatenation performs robustly, while more complex attention-based conditioning did not yield further improvements. The MSE for the U-Net architecture with attention-based conditioning can be compared with DiffusionLLM in A.1. and A.3. in the table 5.

| Method | ETTh1 | | ETTh2 | | ETTm1 | | ETTm2 | | Weather | | ECL | |
|---|---|---|---|---|---|---|---|---|---|---|---|---|
| | MSE | MAE | MSE | MAE | MSE | MAE | MSE | MAE | MSE | MAE | MSE | MAE |
| Diffusion-LLM (Ours) | 0.427 | 0.446 | 0.387 | 0.425 | 0.376 | 0.399 | 0.334 | 0.369 | 0.304 | 0.329 | 0.200 | 0.303 |
| LDM4TS (Ruan et al., 2025) | 0.443 | 0.454 | 0.387 | 0.427 | 0.352 | 0.387 | 0.333 | 0.380 | 0.245 | 0.283 | 0.199 | 0.299 |
| Time-VLM (Zhong et al., 2025) | 0.405 | 0.420 | 0.341 | 0.391 | 0.347 | 0.377 | 0.248 | 0.311 | 0.224 | 0.263 | 0.172 | 0.273 |
| GPT4TS (Zhou et al., 2023a) | 0.465 | 0.455 | 0.381 | 0.412 | 0.388 | 0.403 | 0.284 | 0.339 | 0.237 | 0.270 | 0.167 | 0.263 |
| DLinear (Zeng et al., 2023) | 0.422 | 0.437 | 0.431 | 0.446 | 0.357 | 0.378 | 0.267 | 0.333 | 0.248 | 0.300 | 0.166 | 0.263 |
| PatchTST (Nie et al., 2023) | 0.413 | 0.430 | 0.330 | 0.379 | 0.351 | 0.380 | 0.255 | 0.315 | 0.225 | 0.264 | 0.161 | 0.252 |
| TimesNet (Wu et al., 2023) | 0.458 | 0.450 | 0.414 | 0.427 | 0.400 | 0.406 | 0.291 | 0.333 | 0.259 | 0.287 | 0.192 | 0.295 |
| FEDformer (Zhou et al., 2022) | 0.440 | 0.460 | 0.437 | 0.449 | 0.448 | 0.452 | 0.305 | 0.349 | 0.309 | 0.360 | 0.214 | 0.327 |
| Autoformer (Wu et al., 2021) | 0.496 | 0.487 | 0.450 | 0.459 | 0.588 | 0.517 | 0.327 | 0.371 | 0.338 | 0.382 | 0.227 | 0.338 |
| Stationary (Liu et al., 2022) | 0.570 | 0.537 | 0.526 | 0.516 | 0.481 | 0.456 | 0.306 | 0.347 | 0.288 | 0.314 | 0.193 | 0.296 |
| ETSformer (Woo et al., 2023) | 0.542 | 0.510 | 0.439 | 0.452 | 0.429 | 0.425 | 0.293 | 0.342 | 0.271 | 0.334 | 0.208 | 0.323 |
| LightTS (Zhang et al., 2022) | 0.491 | 0.479 | 0.602 | 0.543 | 0.435 | 0.437 | 0.409 | 0.436 | 0.261 | 0.312 | 0.229 | 0.329 |
| Informer (Zhou et al., 2021) | 1.040 | 0.795 | 4.431 | 1.729 | 0.961 | 0.734 | 1.410 | 0.810 | 0.634 | 0.548 | 0.311 | 0.397 |
| Reformer (Kitaev et al., 2020) | 1.029 | 0.805 | 6.736 | 2.191 | 0.799 | 0.671 | 1.479 | 0.915 | 0.803 | 0.656 | 0.338 | 0.422 |

Table 3: Long-term forecasting results. Each cell shows (MSE, MAE) for a given dataset and method. Results are averaged over four forecasting horizons: $H \in \{96, 192, 336, 720\}$. Lower values (also indicated by darker shade) is better. Even though the main focus and performance gain of our method is as a regularization method for ultra-long-term time series forecasting and data-scarcity scenarios over correponding LLM-only method, it remains competitive with general long-term forecasting baselines.

**Channel Independence and Feature Conditioning:** To investigate whether the channel independence assumption introduced for transformer-based forecasting models (Nie et al., 2023) still holds for our framework with DDPM, we used class conditioning by concatenating a feature ID embedding with the input condition. This modification led to a slight degradation in performance (A.1. and A.2. in the table 5.), suggesting that DDPMs may be sensitive to such conditioning and benefit more from shared latent representations than from explicit feature-wise separation.

**Encoder Sharing and DDPM Contribution:** We conducted ablations to isolate the contribution of the DDPM component and the additional impact of encoder sharing. Adding the DDPM with separate encoders for lookback and forecast windows resulted in a performance gain of approximately 10.81% over the baseline for ultra-long-term forecasting of 2048 timesteps on ETTh1. With a shared encoder for both windows, the DDPM shows additional improvement, with a further reduction in MSE of 12.48% (A.1., B.1. and B.2. in the table 5.) This improvement can be attributed to better semantic alignment in the LLM's embedding space when using a shared encoder, which facilitates more effective distribution learning by the DDPM and enhances overall performance. To analyze the DDPM contribution further, We also added a plot (Figure 4 in Appendix) to show the effect of the regularization weight $\lambda$ on the model performance. The model performs best when $\lambda$ is 1, i.e. when the LLM and DDPM contributes equally to the learning process.

**Efficiency Analysis:** While the model reprogramming approach used by us is more efficient than LLM training or finetuning as shown by Jin et al. (2024), we also analyze the computation overhead introduced to the baseline framework due to the diffusion module. We report training time, GPU memory usage, trainable parameters, and training iteration speed for `Diffusion-LLM` versus TimeLLM in the worst-case ultra-long-term setting with the largest forecast window (2048) in Table 8 in Appendix section A.3. `Diffusion-LLM` introduces minimal additional cost (only 1.82% more GPU memory, 11.54% additional trainable params and 0.39% slower training) compared to TimeLLM, confirming that the added DDPM regularization does not compromise on efficiency. During inference, the diffusion module is not used, ensuring that the inference speed does not get additional overhead over the baseline.

## 6 CONCLUSION

In this work, we introduced `Diffusion-LLM`, a low-overhead but powerful extension to LLM-based time series forecasting frameworks that integrates a conditional diffusion model for distributional regularization. Our method improves performance in ultra-long-term forecasting and few-

shot learning scenarios, where uncertainty and data scarcity pose major challenges. By modeling the conditional distribution of future representations in the shared embedding space, `Diffusion-LLM` enhances the LLM's ability to reason over long horizons and generalize from limited data.

Importantly, our approach introduces only a minimal number of additional trainable parameters relative to the frozen LLM backbone, preserving the efficiency and scalability of the original framework. During inference, the diffusion module is not used, ensuring that the prediction speed remains as fast as the baseline LLM-based framework.

Looking forward, we see several promising directions for future research. First, exploring more expressive or adaptive reprogramming strategies could further improve the alignment between time series and language embedding spaces. Second, there is scope to investigate the role of Diffusion regularizer as an enhancement to the embedding space of other LLM-based and non-LLM models to improve generalizability for any time series model for ultra-long-term forecasting and to analyze the interpretable embedding space changes in LLM-based forecasting baselines for time series reasoning in natural language. Third, incorporating the diffusion model directly into the generation process rather than using it solely for regularization may lead to further gains. Future work could explore using LLMs for temporal encoding combined with DDPM for sequence-by-sequence conditional generation and uncertainty estimation, for example by leveraging DDPM to generate multiple plausible trajectories or estimate predictive variance. Fourth, the current framework can also be extended for uncertainty estimation with a probabilistic head extension instead of point prediction or multiple predictions via dropout. Finally, extending this framework to handle more modalities could broaden its applicability to a wider range of real-world forecasting tasks.

Our `Diffusion-LLM` offers a principled and effective enhancement to time series LLMs, combining the strengths of probabilistic modeling and pretrained language models in a unified framework without loss of efficiency of the original methods.

## 7 REPRODUCIBILITY STATEMENT

We have made extensive efforts to ensure the reproducibility of our work. All implementation details, including model architecture, training procedures, and evaluation protocols are elaborated in the main paper and the Appendix. Hyperparameter configurations for both the LLM and DDPM components are provided in structured tables (table 6 and table 7) within the supplementary materials. Additionally, we include dataset descriptions (4) and preprocessing steps to facilitate replication. An anonymous link to the full source code repository is provided in the subsection A.3, enabling researchers to reproduce our experiments and results with minimal setup.

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

# A APPENDIX

## A.1 DATASET DETAILS

We evaluate **Diffusion-LLM** on six widely-used benchmark datasets for long-term time series forecasting. These datasets span multiple domains, including energy, weather, and offer a diverse testbed for assessing the performance and generalization of our method.

- **ETTm1 and ETTm2**: These datasets are derived from the Electricity Transformer Temperature (ETT) dataset. ETTm1 and ETTm2 contain measurements sampled every 15 minutes, with seven features including oil temperature and load.

- **ETTh1 and ETTh2**: These datasets also comes from the ETT collection but are sampled at an hourly resolution. Like ETTm1 and ETTm2, it includes seven variables, capturing environmental and operational characteristics of electric transformers.

- **Weather**: The Weather dataset is sourced from the UCI Machine Learning Repository and contains meteorological data collected from a local weather station. It includes 21 continuous variables (e.g., temperature, humidity, pressure) recorded every 10 minutes.

- **ECL (Electricity Consumption Load)**: This dataset consists of hourly electricity consumption data from 321 clients in Europe.

| Dataset | Dim. | Dataset Size (Train, Val, Test) | Frequency | Domain | Task |
|---------|------|----------------------------------|-----------|--------|------|
| ETTm1 | 7 | (34465, 11521, 11521) | 15 min | Temperature | Long-term Forecasting |
| ETTm2 | 7 | (34465, 11521, 11521) | 15 min | Temperature | Long-term Forecasting |
| ETTh1 | 7 | (8545, 2881, 2881) | 1 hour | Temperature | Long-term Forecasting |
| ETTh2 | 7 | (8545, 2881, 2881) | 1 hour | Temperature | Long-term Forecasting |
| Weather | 21 | (36792, 5271, 10540) | 10 min | Weather | Long-term Forecasting |
| Electricity | 321 | (18317, 2633, 5261) | 1 hour | Electricity | Long-term Forecasting |

Table 4: Overview of datasets used in Diffusion-LLM. Each dataset varies in dimensionality, sampling frequency, and domain. Forecasting horizons are standardized across all datasets.

For all datasets, we follow the standard data preprocessing and splitting protocols used in prior work such as PatchTST and Time-LLM (Available from the library in `https://github.com/thuml/Time-Series-Library/tree/main`). Specifics of the dataset are added in table 4.

## A.2 EVALUATION METRICS

To evaluate model performance on time series forecasting, we adopt two standard regression metrics:

- **Mean Squared Error (MSE)**: This metric computes the average of the squared differences between the predicted values and the ground truth:

$$\text{MSE} = \frac{1}{N} \sum_{i=1}^{N} (y_i - \hat{y}_i)^2$$

  A lower MSE indicates better performance and penalizes larger errors more heavily due to the squared term.

- **Mean Absolute Error (MAE)**: MAE measures the average absolute difference between predictions and actual values:

$$\text{MAE} = \frac{1}{N} \sum_{i=1}^{N} |y_i - \hat{y}_i|$$

  MAE is more robust to outliers compared to MSE and provides an intuitive measure of forecast accuracy.

## A.3 Experiment Details

The anonymized implementation of `Diffusion-LLM` is available at: `https://anonymous.4open.science/r/blabla-FDEE`.

**Model Architecture:**

Our model adopts a denoising diffusion probabilistic modeling (DDPM) framework for time series forecasting. The underlying structure is a lightweight residual multilayer perceptron (MLP). The model consists entirely of fully connected layersand skip connections.

Let $x \in \mathbb{R}^{B \times L \times D}$ denote a batch of input time series, where $B$ is the batch size, $L$ is the sequence length, and $D$ is the input dimensionality. The model maps a noisy input $x_t$ to a denoised prediction $\hat{x}_0$ through the following components:

**Input and Context Projection:**

The input sequence is projected from $D$ to a hidden dimension $H$ via a linear layer. A conditioning signal (e.g., a context window or past data), also of dimension $D$, is mean-pooled over the temporal axis, broadcast to match the sequence length, and projected into the same hidden space. The two are summed along with a time embedding to produce the initial hidden state:

$$h = \text{Linear}_{\text{in}}(x) + \text{Linear}_{\text{cond}}(\text{repeat}(\text{mean}(c))) + \text{TimeEmbedding}(t)$$

**Time Embedding:**

To encode the diffusion timestep $t$, we use a sinusoidal embedding of dimension $H$, similar to positional embeddings in transformers. This embedding is passed through a linear layer and ReLU activation:

$$t_{\text{emb}} = \text{ReLU}(\text{Linear}_{\text{time}}(\text{Sinusoidal}(t)))$$

This time embedding is broadcast across the temporal dimension and added to the hidden state.

**Class Conditioning (Optional):**

The different features in the dataset are used as different classes for the conditional DDPM. Each class is added to the hidden representation at every timestep.

**Residual Blocks:** The hidden representation is passed through two residual blocks, each consisting of a linear layer followed by a GELU activation and residual skip connection:

$$h \leftarrow h + \text{GELU}(\text{Linear}(h))$$

**Output Projection:**

Finally, a linear output layer maps the hidden representation back to the original input dimension:

$$\hat{x}_0 = \text{Linear}_{\text{out}}(h)$$

**Noise Schedule:**

We experiment with two types of noise schedules for the diffusion process:

- **Linear Schedule.** A simple linear beta schedule is defined as:

$$\beta_t = \text{linspace}\left(\frac{1000}{T} \cdot 10^{-4}, \frac{1000}{T} \cdot 0.02, T\right)$$

  where $T$ is the total number of diffusion steps.

- **Cosine Schedule.** We define the cosine schedule over $T$ steps as:

$$\bar{\alpha}_t = \cos^2\left(\frac{(t/T + s)}{1 + s} \cdot \frac{\pi}{2}\right), \quad \beta_t = 1 - \frac{\bar{\alpha}_{t+1}}{\bar{\alpha}_t}$$

  where $s$ is a small constant (e.g., 0.008), and $\beta_t$ is clipped to the range $[0, 0.999]$ for numerical stability.

| Variant | ETTh1-2048 MSE |
|---|---|
| **A.1.** DiffusionLLM | **0.729** |
| **A.2.** DiffusionLLM with Class Conditioning (A2) | 0.746 |
| **A.3.** DiffusionLLM with Complex U-Net & Attention Conditioning | 0.732 |
| **B.1.** DDPM with Separate Lookback and Forecast Encoders | 0.833 |
| **B.2.** Without DDPM (TimeLLM-style baseline) | 0.934 |

Table 5: Ablations on ETTh1 in predicting 2048 steps ahead (MSE reported). Best result highlighted in bold.

| Task-Dataset | Text Prototype | Backbone Layers | Input Length $T$ | Patch Dim. $d_m$ | Heads $K$ | FF Dim. $d_{ff}$ | LR$^*$ | Loss | Batch Size | Epochs |
|---|---|---|---|---|---|---|---|---|---|---|
| LTF - ETTh1 | 1000 | 32 | 512 | 16 | 8 | 128 | $10^{-3}$ | MSE | 16 | 50 |
| LTF - ETTh2 | 1000 | 32 | 512 | 16 | 8 | 128 | $10^{-3}$ | MSE | 16 | 50 |
| LTF - ETTm1 | 1000 | 32 | 512 | 16 | 8 | 128 | $10^{-3}$ | MSE | 16 | 100 |
| LTF - ETTm2 | 1000 | 32 | 512 | 16 | 8 | 128 | $10^{-3}$ | MSE | 16 | 100 |
| LTF - Weather | 1000 | 32 | 512 | 16 | 8 | 128 | $10^{-2}$ | MSE | 64 | 100 |
| LTF - ECL | 1000 | 32 | 512 | 16 | 8 | 32 | $10^{-2}$ | MSE | 128 | 100 |

Table 6: LLM hyperparameters used for each dataset in Diffusion-LLM. All models use the same LLaMA-7B backbone with frozen weights.

| Hyperparameter | Value / Description |
|---|---|
| input_dim | 4096 (Dimensionality of input time series patches) |
| hidden_dim | 512 (Hidden layer size used throughout the DDPM model) |
| time_emb_dim | 512 (Dimensionality of sinusoidal time embedding) |
| num_classes | 0 (No class conditioning used in final version) |
| residual_blocks | 2 (Number of residual blocks in the DDPM architecture) |
| activation | GELU (Activation function used in residual blocks) |
| output_proj | Linear (Final layer to project hidden state back to input dimension) |
| timesteps | 1000 (Total number of diffusion steps) |
| beta_schedule | cosine (Noise schedule used for diffusion process) |
| sampling_timesteps | 1000 (Number of steps used during sampling) |
| objective | pred_noise (Training objective: predict added noise) |
| loss_function | MSE (Loss computed between predicted and target noise) |
| self_conditioning | False (Optional technique to improve sample quality; not used) |
| parameter_count | $\sim$7M (Approximate number of parameters added by DDPM) |

Table 7: DDPM hyperparameters used in Diffusion-LLM. These settings are shared across all datasets.

To avoid underestimating our baseline, for the LLM part, we use the same hyperparameters as Jin et al. (2024) apart from Weather and Electricity dataset where we use larger batch size of 64 and 128 to accommodate computing time. The hyperparameters are listed in the table 6.

For our DDPM architecture, we use same hyperparameters for all datasets. It is a residual MLP-based backbone with a hidden dimension of 512 throughout. The input and conditioning sequences, each with dimensionality 4096, are projected to the hidden space using separate linear layers. The model includes two residual blocks, each with a single linear layer followed by GELU activation and skip connection. A sinusoidal time embedding of size 512 is used, followed by a linear projection to match the hidden dimension. The output is projected back to the original input dimension via a final linear layer. Overall, the model contains six main linear layers, all operating at the hidden size of 512. The DDPM model adds only approximately 7 M parameters. Further, adding the condition into the DDPM model in different scenarios for different datasets always yielded similar results with 1-2 percent deviation only in either direction, hence in the final version, we have not used the class conditioning. The DDPM hyperparameters are listed in the table 7.

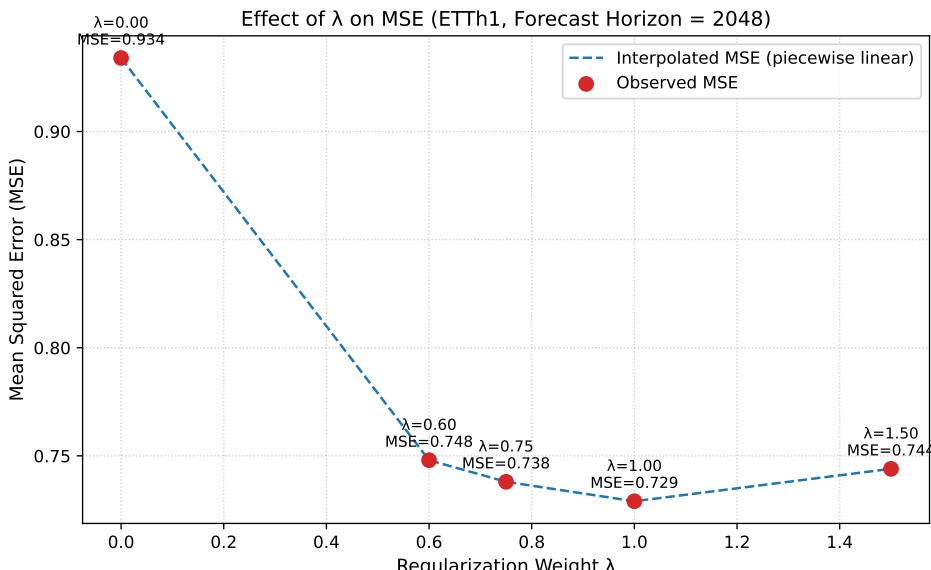

Figure 4: Impact of regularization weight ($\lambda$) on forecasting performance (MSE) for ETTh1 dataset with a 2048-step horizon. The plot shows that $\lambda = 1$ achieves the best performance (MSE = 0.729), indicating that an equal contribution from the forecasting loss and the diffusion-based regularization provides optimal balance. Smaller $\lambda$ values (e.g., 0 for TImeLLM or 0.6) under-regularize the embedding space, limiting the benefit of distribution-aware alignment, while larger $\lambda$ values (e.g., 1.5) overemphasize the diffusion objective, causing over-regularization and slight performance degradation. This demonstrates the importance of tuning $\lambda$ to balance deterministic forecasting and probabilistic embedding refinement.

| Model | Training Time (GPU-h) | Max GPU Mem Usage (MiB) | Trainable Params (M) | Speed (s/iter) |
|---|---|---|---|---|
| Diffusion-LLM | 6.437 | 33188 | 6.461 | 0.397 |
| TimeLLM | 6.461 | 32592 | 6.437 | 0.395 |

Table 8: Efficiency analysis for ETTh1 forecasting 2048 steps ahead. Training time and resource usage are reported for Diffusion-LLM and TimeLLM.

