# OpenReview forum: "Diffusion-LLM Provides Ultra-Long-Term Time Series Forecasting with Probabilistic Alignment"
_ICLR.cc/2026/Conference — ICLR 2026 Conference Withdrawn Submission_

### Official Review · Reviewer_MftG · 2025-10-16

**Soundness:** 2
**Presentation:** 2
**Contribution:** 1
**Rating:** 2
**Confidence:** 4

**Summary:**

This paper proposes Diffusion-LLM, which integrates a conditional DDPM into LLM-based time series forecasting to improve ultra-long-term predictions. The DDPM models the conditional distribution $p(z_y|z_x)$ in the shared embedding space and acts as a regularizer during training. Experiments on 6 benchmarks show improvements over TimeLLM in ultra-long-horizon (H=1024-2048) and few-shot settings, but demonstrate inconsistent performance on standard long-term forecasting.

**Strengths:**

S1. Addresses Under-Explored Problem: Ultra-long-term forecasting is an important yet under-explored problem.
S2. Reasonable Methodological Design: Using conditional DDPM to model the distribution in embedding space is conceptually sound.
S3. Reproducibility: Hyperparameters, dataset details, and anonymous code are provided in the Appendix, and easy to implement on the BasicTS benchmark.

**Weaknesses:**

**W1. Insufficient Experimental Validation**:

The proposed method was only compared with TimeLLM and TS-specific models before 2024. Comparisons with the competitive/SOTA models of the last two years are essential. Furthermore, except TS-specific models, LM-based models e.g GPT4TS/TimeVLM, and diffusion-based models e.g. Diffusion-TS,  LM+diffusion work e.g. LDM4TS are necessary to compare.

**W2. Limited Novelty**:

The proposed method mainly combines existing components (TimeLLM reprogramming + standard DDPM). The technical contribution is more engineering-oriented than methodologically innovative, lacking theoretical justification for the approach.

**W3. Limited improvements:**

For example, Table 3 shows Diffusion-LLM is worse than PatchTST and DLinear on some dataset. The experiments fail to convincingly demonstrate the method's advantages.

**W4. Presentation Problems:**

There is a lot of white space around the figures and formulas, which gives the impression that the manuscript is under-written.

**W5. Missing Efficiency Study:**

The authors claim their method preserves "efficiency" provide no empirical evidence: no training time, GPU memory, or FLOPs comparison despite claiming minimal overhead. Moreover, DDPM is known to be substantially slower than alternatives like DDIM, yet the paper offers no justification for choosing DDPM. The lack of efficiency analysis and the justification for diffusion/LLM model choices makes the efficiency claims unsubstantiated and raises concerns about practical applicability.

**W6. Results on ILI datasets are missing.**

**Questions:**

Q1: Why does DDPM act as an effective regularizer for LLM-based forecasting? What is the justification for this specific method choice compared to alternatives?

Q2: Why is "trained with DDPM, inferred without" superior to direct LLM training? Is there empirical evidence or ablation study supporting this design choice?

Q3: How does performance vary with λ in Eq. (8)?

Q4: Where is the ablation study mentioned in Section 5? Are the results missing from the tables/figures?

Q5: Can the method provide uncertainty quantification at inference time, given that both DDPM and LLMs have capacity for probabilistic modeling?

---

> ### Author Response · Authors · 2025-11-25
> **Response to Reviewer MftG (Part 1)**
>
> **W1**. Insufficient Experimental Validation:
>
> **Response1**. Thank you for highlighting the additional references. Diffusion specific models generally use much shorter forecasting lengths because of difficulty of modeling temporal relation. This is also true for Diffusion-TS where only very short forecasting length of 36 was used compared to the forecasting length of 2048 used in our experiment. However, LDM4TS uses a vision transfer on the visual representations to encode temporal patterns directly and is used for long-term prediction. We have added the results from LDM4TS and TimeVLM to our comparison in table 3. GPT4TS is already part of the results.
>
> **W2**. Limited Novelty:
>
> **R2**. We appreciate the concern regarding novelty and would like to explain that our contribution goes beyond simply combining TimeLLM reprogramming with a standard DDPM. The core innovation lies in embedding space refinement for multimodal alignment, which is critical for leveraging pretrained LLMs in time series forecasting. Pretrained LLMs operate in a highly structured language embedding space, and their performance depends on the semantic coherence of this space. When time series data is reprogrammed into token-like representations, the alignment between these representations and the LLM’s native word prototypes becomes the bottleneck for effective reasoning. Our method introduces a distribution-aware regularization mechanism that uses DDPM to learn the conditional distribution of forecast-window word prototypes given lookback-window word prototypes. The theoretical idea is to selectively emphasize the most probable word prototypes for describing time series characteristics, thereby reducing noise and improving semantic alignment and giving the LLM a better distribution of word prototypes to choose from. Since forecasting accuracy in frozen LLMs will be strongly correlated with the quality of the shared embedding space, by refining this space through probabilistic modeling, we enable the LLM to capture the time series semantics better in language space without altering its pretrained attention parameters. This is fundamentally different from prior works that either fine-tune LLM weights or use DDPM as a generative forecaster. Our design treats DDPM as an auxiliary learner which is novel in the context of multimodal LLMs for time series.
>
>
> **W3**. Limited improvements:
>
> **R3**. We have discussed about it in our general comment under Performance with Long-Term Data.
>
>
> **W4**. Presentation Problems:
>
> **R4**. Thank you for the feedback. We have made the presentation more concise particularly between consecutive equations and made the figures more symmetric in the revision while adhering to the conference template tables and figures guidelines.
>
> **W5**. Missing Efficiency Study:
>
> **R5**. Thank you for this important point. As mentioned in the main comment under Efficiency Analysis, we have added it in table 5 and discussed in section 5. Regarding DDPM/DDIM, we do not use the diffusion model during inference and since DDIM’s main efficiency comes in sample generation, it is not directly relevant for the current framework.
>
>
> **W6**. Results on ILI datasets are missing.
>
> **R6**. We had initially considered ILI dataset but as noted in the lines 316-318, shorter standard forecasting window of H ∈ {24, 36, 48, 60} and unavailability of enough data for ultra-long forecasting make ILI unsuitable for our evaluation.

---

> ### Author Response · Authors · 2025-11-25
> **Response to Reviewer MftG (Part 2)**
>
> **Q1**: Why does DDPM act as an effective regularizer for LLM-based forecasting? What is the justification for this specific method choice compared to alternatives?
>
> **A1**: Thank you for allowing us to clarify our methodological motivation. Following from W2, the goal is to refine the shared embedding space using the understanding of the forecast window’s text prototype conditioned on the lookback window’s text prototype. So, we chose a probabilistic model like DDPM over other generative models (that avoids GAN’s training complexity or VAE’s parametric latent).
>
>
> **Q2**: Why is "trained with DDPM, inferred without" superior to direct LLM training? Is there empirical evidence or ablation study supporting this design choice?
>
> **A2**: Thank you for raising this point. We do not generate our output in an autoregressive manner. The goal in the current framework is to refine model reprogramming for LLM and our design does not require the DDPM in inference. This also removes the overhead of the slow denoising/sample generation process. Our model reprogramming framework has total 6.675 B parameters out of which only 68M parameters are trainable which is only 1.02% of the number of params to be trained for full LLM training. TimeLLM paper shows that model reprogramming is more efficient than LLM finetuning too.
>
> **Q3**: How does performance vary with λ in Eq. (8)?
>
> **A3**: Thank you for this important question regarding the training weightage between LLM and DDPM. We empirically observed that the model performs the best with equal contribution from the two, i.e. for a lambda value of 1. We have added this observation in lines 467-470 in our revision and an example illustration for the performance variation with lambda in figure 4.
>
> **Q4**: Where is the ablation study mentioned in Section 5? Are the results missing from the tables/figures?
>
> **A4**: Thank you for pointing this out. We have added the table and discussion as mentioned in general comment under Tables Corresponding to the Model Analysis in Section 5.
>
> **Q5**: Can the method provide uncertainty quantification at inference time, given that both DDPM and LLMs have capacity for probabilistic modeling?
>
> **A5**: Thank you for this very interesting point of discussion. In the current framework, we are using a linear layer to directly extract point forecasts. It can be extended for prediction uncertainty estimation in a few ways, for example, with multiple predictions along with dropput, or a probabilistic head with modified loss for variance estimation. For DDPM, this can only be done with sample generation at inference. We have added more elaboration on this in conclusion lines 500-505. However, as also can be seen from the tables 1 and 2, Diffusion-LLM MSE and MAE is much more stable than the baseline TimeLLM with lower variances in the results for almost all scenarios in long, ultra-long and few-shot scenarios.
>
> Thank you again for your constructive feedback and the questions. These have enabled us to improve the paper by modifying the revision with more results, efficiency information, future research directions as well as providing more explanations into the model’s strengths and limitation.

---

> > ### Comment · Reviewer_MftG · 2025-11-27
> >
> > Thank you for your responses. My concerns have been partially addressed. I suggest presenting the new content that was added to the revised submission in your reply, as R1 and R5 still lack experimental data to be presented. However, I will raise my score to 4 in recognition of your hard work.

---

### Official Review · Reviewer_HHxd · 2025-10-23

**Soundness:** 2
**Presentation:** 3
**Contribution:** 2
**Rating:** 4
**Confidence:** 5

**Summary:**

Proposes Diffusion-LLM, a framework integrating conditional Denoising Diffusion Probabilistic Model (DDPM) into LLM-based time series forecasting for probabilistic alignment and distribution-aware regularization.

**Strengths:**

- Targets LLM’s time series flaws (poor probabilistic modeling, weak alignment) via DDPM, with gains in ultra-long-term/few-shot scenarios.
- Lightweight DDPM avoids heavy params; inference speed matches baseline LLMs.

**Weaknesses:**

- Incremental innovation: relies on existing components (LLaMA, TimeLLM’s reprogramming, DDPM) with limited novel design.
- High training compute (A100/H100) but no quantitative training time/energy vs. baselines.
- Underperforms non-LLM baselines (e.g., PatchTST) in standard long-term forecasting.

**Questions:**

-  What unique designs distinguish it from prior generative-regularized LLMs for other modalities?
- Can training compute (GPU hours/memory) be quantified vs. TimeLLM/non-LLM baselines?
- Why not integrate DDPM into inference (e.g., probabilistic forecasting), and would this improve performance?
- For underperforming datasets (e.g., ETTm2, ECL), do conflicting LLM-DDPM objectives cause issues?

---

> ### Author Response · Authors · 2025-11-25
> **Response to Reviewer HHxd**
>
> **Q1**: What unique designs distinguish it from prior generative-regularized LLMs for other modalities?
>
> **A1**: Thank you for allowing us to clarify more about our methodological novelty. The main combination of diffusion and multimodal LLMs in literature are done for the multimodal image generation models where Diffusion models are used to extract text features from LLM controllers directly or through learnable connectors and generate the output [1]. Our design is completely different as we use the diffusion model to learn the conditional distribution in the word embedding space. It learns the relation between the word prototypes representative of the lookback window and the word prototypes representative of the forecast window, thus refining the density of the word prototypes and enabling the pretrained frozen LLM to pay appropriate ‘attention’ to the timeseries-relevant word prototypes. To the best of our knowledge, there is no similar work on generative models as probabilistic regularizer in the shared embedding space for multimodal LLMs.
>
>
>
>
> **Q2**: Can training compute (GPU hours/memory) be quantified vs. TimeLLM/non-LLM baselines?
>
> **A2**: Thank you for the question. As stated in the general comment under Efficiency Analysis, we have added the table with these details in our revision.
>
>
>
>
> **Q3**: Why not integrate DDPM into inference (e.g., probabilistic forecasting), and would this improve performance?
>
> **A3**: Since this is an interesting and significant point, we have addressed this in the general review comment under DDPM in inference.
>
>
>
>
> **Q4**: For underperforming datasets (e.g., ETTm2, ECL), do conflicting LLM-DDPM objectives cause issues?
>
> **A4**: Thank you for this very important question. Yes, the short-term patterns learning and long-term uncertainty modelling offers conflict causing over-regularization for datasets that are easier to forecast. We have elaborated on this more in our general comment under Performance with long-term data.
>
> Thank you again for your constructive feedback and the questions. These have enabled us to improve the paper by modifying the revision for novelty and uniqueness of our method, efficiency information, future research directions as well as providing more explanations into the model’s strengths and limitation.
>
> [1]. Hu, Xiwei, et al. "Ella: Equip diffusion models with llm for enhanced semantic alignment." arXiv preprint arXiv:2403.05135 (2024).
> [2] Yongliang Shen, Kaitao Song, Xu Tan, Dongsheng Li, Weiming Lu, and Yueting Zhuang. 2023. HuggingGPT: solving AI tasks with chatgpt and its friends in hugging face. In Proceedings of the 37th International Conference on Neural Information Processing Systems (NIPS '23).

---

> > ### Comment · Reviewer_HHxd · 2025-11-27
> > **Official Comment of Reviewer HHxd**
> >
> > Thank you for your reply. However, I will keep my score.

---

### Official Review · Reviewer_rn7y · 2025-10-24

**Soundness:** 2
**Presentation:** 1
**Contribution:** 2
**Rating:** 2
**Confidence:** 5

**Summary:**

The paper targets LLM forecasters’ “regress-to-mean” and weak probabilistic alignment, proposing **Diffusion-LLM**: reprogram time series into an LLM token space and attach a **conditional DDPM** that learns $(p(z_y\mid z_x))$ to regularize the LLM with distribution-aware signals. Training minimizes a joint objective $(L_{\text{forecast}}+\lambda L_{\text{ddpm}})$; only the frozen-backbone LLM path is used at inference, keeping latency unchanged. Experiments on several standard benchmarks demonstrate consistent improvements over baselines, especially in ultra-long-term (1024–2048 steps) and few-shot forecasting tasks.

**Strengths:**

1. The paper is well-structured and easy to follow.
2. The proposed method shows relatively strong performance under the ultra long-term forecasting setting.

**Weaknesses:**

1. Lines 38–39 claim that many real-world scenarios require predictions far beyond the typical forecasting range. However, the authors do not provide concrete examples of such domains. **It is unclear why ultra long-term forecasting is necessary when sufficient historical data exists, as many practical applications—especially in finance—still focus on single-step or short-horizon predictions.**
2. Lines 57–59 argue that LLMs tend to predict the mean and rely more on nearby points when forecasting long-term. **Yet this issue is common across many deep learning time series models due to their training paradigms, not unique to LLMs.** Hence, the motivation for combining LLMs with diffusion models is not well justified.
3. Following the previous point, if one uses a well-designed time series model such as PatchTST [1] or iTransformer [2] together with the proposed DDPM module, the performance might surpass that of LLM-based models. If so, the role of LLMs here seems limited.
4. Table 1 shows that under standard forecasting horizons, the proposed model performs worse than TimeLLM [3]. This further raises concerns that the method offers little advantage in common forecasting scenarios, which are far more prevalent than ultra long-term ones.
5. Section 5 (lines 432–463) contains only analysis but no experimental results, which is unusual and weakens the empirical evidence.
6. The citation format is used incorrectly throughout. When citations are not part of the sentence structure, the author should use `\citep` instead of `\cite`.

[1] A Time Series is Worth 64 Words: Long-term Forecasting with Transformers

[2] iTransformer: Inverted Transformers Are Effective for Time Series Forecasting

[3] Time-LLM: Time Series Forecasting by Reprogramming Large Language Models

**Questions:**

See **Weakness**.

---

> ### Author Response · Authors · 2025-11-25
> **Response to Reviewer rn7y**
>
> **Q1**: Lines 38–39 claim that many real-world scenarios require predictions far beyond the typical forecasting range. However, the authors do not provide concrete examples of such domains. It is unclear why ultra long-term forecasting is necessary...
>
> **A1**: While it is true that financial applications focus on short-horizon, ultra-long-forecasting is required in other domains like Energy sector, climate science, vehicle industry etc. Prediction needs in energy demand forecasting can range anywhere between hours to years [1]. Lack of historical data is a factor in use cases like long-term Battery health forecasting from early age data [2]. Thank you for pointing out the missing motivational clarity, we have now included it in detail in the revised Introduction section’s line 037-044.
>
> **Q2**: Lines 57–59 argue that LLMs tend to predict the mean and rely more on nearby points when forecasting long-term. Yet this issue is common across many deep learning time series models due to their training paradigms, not unique to LLMs. Hence, the motivation for combining LLMs with diffusion models is not well justified.
> **Q3**: Following the previous point, if one uses a well-designed time series model such as PatchTST [1] or iTransformer [2] together with the proposed DDPM module, the performance might surpass that of LLM-based models. If so, the role of LLMs here seems limited.
>
> **A2 & 3**: Thank you for these very interesting points as it allows us to clarify more on the motivation of our approach. Our motivation behind combining DDPM and LLM is two-fold:
>
> a. As mentioned in Introduction, LLMs have strong reasoning abilities due to their vast pretrained knowledge. To use that for time series, it is essential to perform the multimodal alignment as LLMs are trained in language embedding space. Since they are relying on their pretrained attention modules, the key to good performance is a good embedding space.
>
> b. Hence, we use DDPM to learn the distribution in the embedding space and improve this shared embedding space of time series and language. This can be interpreted as focusing more on the word prototypes that can describe time series characteristics better.
> As mentioned in conclusion, we present our method as an add-on improvement over existing LLM-based methods. DDPM works on the language embedding space and we use TimeLLM as our example baseline. While this research is to advance the state of the research for multimodal alignment in time series, models like PatchTST are specialized time series models giving a non-interpretable time series embedding space but it is possible that in a separate context, this method can improve PatchTST too to make our method even more generalizable. In an extended version, we will investigate the effect of Diffusion regularizer in the embedding space of other non-LLM models to compare its benefits to improve any general time series model for ultra-long-term forecasting and also look into the interpretable embedding space changes for original LLM-based forecasting baselines for time series reasoning. We have added this in our Conclusion section(496-499) in the revision.
>
>
> **Q4**: Table 1 shows that under standard forecasting horizons, the proposed model performs worse than TimeLLM [3]. This further raises concerns that the method offers little advantage in common forecasting scenarios, which are far more prevalent than ultra long-term ones.
>
> **A4**: Thank you for this important question. We have discussed it in our general answer under Performance with Long-Term Data. Regarding our motivation for a model with specialization advantage in ultra-long-term forecasting, while long term forecasting is a well-researched problem, ultra long forecasting is underexplored, providing little alternative in the domains where very long forecasting is essential.
>
>
> **Q5**: Section 5 (lines 432–463) contains only analysis but no experimental results, which is unusual and weakens the empirical evidence.
>
> **A5**: As discussed in the general comment under Tables Corresponding to the Model Analysis in Section 5, we have updated our manuscript.
>
> **Q6**: The citation format is used incorrectly throughout. When citations are not part of the sentence structure, the author should use \citep instead of \cite.
>
> **A6**: Thank you for pointing this out. We have corrected the citation format in the revision.
>
> Thank you again for your time and constructive feedback. This has enabled us to make several improvements in the revised version including the motivation, future research direction, as well as reinforcing the strengths and acknowledging the limitations of our method.
>
> [1] J. Rodrigues Dos Reis et al., "Medium and Long Term Energy Forecasting Methods: A Literature Review," in IEEE Access
> [2] Tingkai Li, Zihao Zhou, Adam Thelen, David A. Howey, Chao Hu.
> Predicting battery lifetime under varying usage conditions from early aging data

---

> > ### Comment · Reviewer_rn7y · 2025-11-26
> >
> > I can understand, to some extent, the idea of using diffusion as an alignment or refinement mechanism. However, I am still not fully convinced by the task setting itself, particularly the claim that the ultra-long-term forecasting results are truly compelling. A key concern is that, under both the long-term and ultra-long-term experimental settings, the authors only compare against TimeLLM. In practice, I believe that other time series models such as PatchTST, iTransformer, or even models specifically optimized for ultra-long-term forecasting, if directly trained on this task, would likely achieve competitive or even better performance. (Sorry that I did not express this point clearly in my previous comment.)
> >
> > While I agree that LLMs possess generalized pretrained knowledge, which is indeed valuable, I remain skeptical about its necessity in ultra-long-term forecasting scenarios. Compared to short-term and conventional long-term forecasting, ultra long-term scenarios are far less common and, in my view, require more specialized domain knowledge rather than generalized knowledge learned from broad pretraining.
> >
> > In summary, I appreciate the authors’ serious and constructive attitude toward the review process. However, I believe this work still requires more careful refinement, particularly regarding the fundamental motivation and positioning of the task. Therefore, I will maintain my original score.

---

### Official Review · Reviewer_VCGC · 2025-10-31

**Soundness:** 2
**Presentation:** 3
**Contribution:** 4
**Rating:** 4
**Confidence:** 4

**Summary:**

In this paper the authors build upon the work of TimeLLM to study multimodal time series forecasting with existing LLM backbones. Their main contribution is the additional alignment of the encoded target time series into the LLM embedding space by introducing a denoising diffusion probabilistic modeling task between the input and the target time series as a regularization during model training. They empirically show that this added alignment helps with long term forecasting and forecasting in data scarce regimes.

**Strengths:**

The paper studies an interesting and legit question of time series tokenizer at the presence of the requirement of LLM alignment. The proposed method for aligning the target time series embeddings is clean. Paper writing is good.

**Weaknesses:**

The foundation of the research is not well established, which makes it hard to assess the significance and the generalization of the proposed work. In particular: (1) It is not clear that the base method, TimeLLM, is a good approach for tacking multimodal time series forecasting. (2) The selected benchmark datasets, which have been pervasive (overused) in time series research, might not be the best multimodal test cases. See questions.

**Questions:**

1. My most concern is about the benchmarking datasets. Though overly used by time series research, these datasets might not provide enough diversity in terms of the text prompts. In addition, based on the PatchTST performance in Figure 3, they are not essentially multimodal forecasting problems. What's your justification of the expectation that the proposed method can generalize to other real world usecases?

2. The proposed method seems to be not improving or strictly hurting the performance for the long term tasks with full data, (see Table 1, take into consideration the CIs).

3. How does the trained DDPM stand as a unimodal forecaster?

---

> ### Author Response · Authors · 2025-11-25
> **Response to Reviewer VCGC**
>
> **Q1**: My most concern is about the benchmarking datasets. Though overly used by time series research, these datasets might not provide enough diversity in terms of the text prompts. In addition, based on the PatchTST performance in Figure 3, they are not essentially multimodal forecasting problems. What's your justification of the expectation that the proposed method can generalize to other real world usecases?
>
> **A1**: We understand the concern about the diverse domain knowledge to be transferred for real world usecases. However, the text prompt we use for this method is a combination of (domain and dataset information + task instruction + simple derived statistical information like average, minimum, maximum etc. from the training window). The time series forecasting problem, on its own, is not expected to be a multimodal problem. In fact, previous works have directly used time series patches as tokens and used them as inputs to LLMs with embedding [1]. However, we recognize the modality gap between time series and language and hence formulate the problem in a multimodal way.
>
> For the first part, i.e. domain and dataset information, we only use extremely simple information to ensure that the method is not unfairly dependent on the existing knowledge of the dataset. For example, the description used for Weather dataset is simply “Weather is recorded every 10 minutes for the 2020 whole year, which contains 21 meteorological indicators, such as air temperature, humidity, etc.”. We believe this amount of information is reasonable to be expected for real world dataset also. The second part, i.e. task instruction is: Predict the next 1024/2048 timesteps based on the previous 512 timesteps. And the last part, i.e. statistical information is calculated at training time with simple mathematical operator functions. Hence, even for real world dataset, the second and third part remains unchanged and providing additional domain information can be handled trivially.
>
> Other models that handle the forecasting problem in a multimodal way also use these datasets in a similar way for evaluation[2][3]. However, we acknowledge that the prompting strategy should be elaborated more in the paper to clarify this doubt for the reader. We thank you for drawing our attention to addressing this point and have now added a detailed description for this in our revised manuscript in the “Methodology” section component A lines 193-198.
>
>
>
>
> **Q2**: The proposed method seems to be not improving or strictly hurting the performance for the long term tasks with full data, (see Table 1, take into consideration the CIs).
>
> **A2**: Thank you for this important point. We have addressed this in our general comment on Performance with Long-Term Data.
>
>
>
>
> **Q3**: How does the trained DDPM stand as a unimodal forecaster?
>
> **A3**: Thank you for raising this important point. DDPM does not perform well as a unimodal forecaster for general long term time series forecasting since apart from the probabilistic distribution of the forecast window, it also needs to capture the temporal relationship in the lookback window and generally that is done with an autoregressive model separately. We have added a more detailed description and future research direction with DDPM as time series forecaster in our general comment on DDPM in Inference.
>
>
> Thank you again for your time and the constructive feedback. This has enabled us to make several improvements in the revised version e.g. adding more methodical clarifications, reinforcing the strengths and acknowledging the limitations of our method, and future research directions.
>
>
> [1] Tian Zhou, Peisong Niu, Xue Wang, Liang Sun, and Rong Jin. 2023. One fits all: power general time series analysis by pretrained LM. In Proceedings of the 37th International Conference on Neural Information Processing Systems (NIPS '23). Curran Associates Inc., Red Hook, NY, USA, Article 1877, 43322–43355.
> [2] Vision-Enhanced Time Series Forecasting via Latent Diffusion Models.
> Weilin Ruan, Siru Zhong, Haomin Wen, Yuxuan Liang
> [3] Time-VLM: Exploring Multimodal Vision-Language Models for Augmented Time Series Forecasting. Siru Zhong, Weilin Ruan, Ming Jin, Huan Li, Qingsong Wen, Yuxuan Liang. ICML 2025

---

### Author Response · Authors · 2025-11-25
**General response to all reviewers**

General response to all reviewers:

We sincerely thank all the reviewers for their time and the constructive and detailed feedback. We are quite encouraged by the recognition that the paper addresses under-explored, significant, and interesting problems (Reviewers VCGC, MftG), that the paper writing is good (Reviewers VCGC, rn7y), and that the proposed approach is methodically clean, conceptually sound and reproducible with strong performance in ultra-long-term forecasting (Reviewer VCGC, rn7y, HHxd, and MftG). These positive observations reinforce the motivation, relevance, and methodology of our work.

We now address the key questions and recurring suggestions raised by multiple reviewers through this general response and appreciate the opportunity to further refine and strengthen the manuscript. We have carefully revised the paper, and the edits have been highlighted in BLUE. While we discuss the major comments and revisions here, we also address all the remaining comments and revisions to each reviewer individually.

1.	**Performance with Long-Term Data** (Reviewers VCGC, rn7y, HHxd, MftG):
The diffusion model’s ability to capture probability distributions makes DDPM-based regularization most effective in scenarios with high uncertainty. Consequently, our method is optimized for uncertainty-heavy, low-information regimes such as ultra-long-term forecasting and few-shot learning that involves weaker correlations with recent history and significantly higher uncertainty. In these cases, the DDPM regularizer is particularly valuable because it models the full conditional distribution of future trajectories and maximizes it for likelihood. Hence, the joint loss function combining forecasting and diffusion objectives introduces an optimization trade-off. While this improves robustness in high-uncertainty settings, it can slightly reduce point prediction accuracy for shorter horizons where deterministic patterns dominate.
This trade-off is empirically evident in Tables 1 and 2. Diffusion-LLM delivers the greatest benefits on more challenging datasets, which exhibit higher baseline MSE. Even for relatively easier datasets with lower MSE, as forecasting windows lengthen and training data becomes scarce, the advantages of Diffusion-LLM become increasingly clear. In these difficult and uncertain scenarios, our method consistently outperforms the baseline, reinforcing its design focus on robustness under uncertainty.
We have also incorporated this elaboration at the end of the Results section (lines 403-413) in the revised manuscript to clearly explain the optimization trade-off and its implications for different forecasting regimes.

2.	**Tables Corresponding to the Model Analysis in Section 5** (Reviewers rn7y, MftG):
We have added the experimental results corresponding to our model analysis and ablation studies in table 5 in appendix and discussed the results in section 5.

3.	**Efficiency Analysis** (Reviewers HHxd, MftG):
We have provided the information and comparative study on training time, number of parameters, training speed and memory requirements in table 8 in appendix and added a small paragraph Efficiency Analysis to discuss the findings in section 5.

4.	**DDPM in Inference** (Reviewers VCGC, HHxd, MftG):
While DDPM could theoretically be integrated into inference for probabilistic forecasting, DDPMs alone generally cannot effectively capture temporal dependencies in time series. They are typically paired with autoregressive models, and the framework suffer from error accumulation and slow generation for long horizons [1]. Our work instead leverages DDPM as a training-time regularizer to improve multimodal alignment between time series and language embeddings in LLMs. Future work could explore using LLMs for temporal encoding combined with DDPM for sequence-by-sequence conditional generation and uncertainty estimation, but this requires substantial architectural changes and is not a direct extension of our current framework. However, we acknowledge that this is a potentially significant stream of future research and discuss it in further detail in the conclusion.

[1]. Yuan, X.; Qiao, Y. Diffusion-TS: Interpretable Diffusion for General Time Series Generation. In Proceedings of The Twelfth International Conference on Learning Representations, 2024. Available online: https://openreview.net/forum?id=4h1apFjO99

---

### Note · Authors · 2026-01-19

I have read and agree with the venue's withdrawal policy on behalf of myself and my co-authors.